# Comparison of Genotype II African Swine Fever Virus Strain SY18 Challenge Models

**DOI:** 10.3390/v15040858

**Published:** 2023-03-27

**Authors:** Xintao Zhou, Jiaqi Fan, Xiaopan Guo, Teng Chen, Jinjin Yang, Yanyan Zhang, Lijuan Mi, Fei Zhang, Faming Miao, Min Li, Rongliang Hu

**Affiliations:** 1College of Life Sciences, Ningxia University, Yinchuan 750021, China; 2Key Laboratory of Prevention & Control for African Swine Fever and Other Major Pig Diseases, Ministry of Agriculture and Rural Affairs, Changchun 130122, China; 3Changchun Veterinary Research Institute, Chinese Academy of Agricultural Sciences, Changchun 130122, China

**Keywords:** ASFV strain SY18, administration routes, major clinical symptoms, challenge models, intraoral, intranasal, intramuscular

## Abstract

African swine fever (ASF) is a viral haemorrhagic disease found in domestic and wild boars caused by the African swine fever virus (ASFV). A highly virulent strain was used to evaluate the efficacy of newly developed vaccine candidates. The ASFV strain SY18 was isolated from the first ASF case in China and is virulent in pigs of all ages. To evaluate the pathogenesis of ASFV SY18 following intraoral (IO) and intranasal (IN) infections, a challenge trial was conducted in landrace pigs, with intramuscular (IM) injection as a control. The results showed that the incubation period of IN administration with 40–1000 50 % tissue culture infective dose (TCID_50_) was 5–8 days, which was not significantly different from that of IM inoculation with 200 TCID_50_. A significantly longer incubation period, 11–15 days, was observed in IO administration with 40–5000 TCID_50_. Clinical features were similar among all infected animals. Symptoms, including high fever (≥40.5 °C), anorexia, depression, and recumbency, were observed. No significant differences were detected in the duration of viral shedding during fever. There was no significant difference in disease outcome, and all animals succumbed to death. This trial showed that IN and IO infections could be used for the efficacy evaluation of an ASF vaccine. The IO infection model, similar to that of natural infection, is highly recommended, especially for the primary screening of candidate vaccine strains or vaccines with relatively weak immune efficacy, such as live vector vaccines and subunit vaccines.

## 1. Introduction

African swine fever virus (ASFV) is the causative agent of African swine fever (ASF), a devastating haemorrhagic disease that affects domestic pigs and wild boars [1]. ASFV is a structurally complex enveloped double-stranded DNA virus that belongs to the genus *Asfivirus*, family *Asfaviridae*, and is currently the only member of this family [2]. There are a total of 24 genotypes according to the variations in 247 amino acids at the C-terminal of the p72 protein [1,3]. Genotypes I and II have spread in recent years, and many variants within each genotype have been identified [4,5,6,7]. The size of the ASFV genome of different field isolates ranges among different isolates, containing 151–167 open reading frames (ORFs) [3,8]. The pathogenicity of different ASFV isolates with varied genome sizes or constitutions are reportedly virulent and avirulent, and they lead to peracute, acute, subacute, chronic, or subclinical infections and clinical outcomes [1,2,9,10]. 

ASF control and outbreak management rely on restricting animal movement and culling infected herds because no vaccine or treatment is available. Live attenuated viruses developed by the genetic manipulation of virulent strains, or naturally attenuated viruses, have been shown to provide vaccinated animals with solid protection against ASF. In 2022, ASFV-G-ΔI177L, one of the gene-deleted ASF vaccine candidates, was approved for commercial use in Vietnam [11,12,13]. However, it raises biosafety concerns related to long-term ASF live vaccine application. With the use of vaccines, ASFV mutations may occur and result in unpredictable consequences, such as virulence reversal, severe side effects, and difficulty in the clearance of the mutated virus. Therefore, ASF vaccines are crucial, yet they remain elusive. Biosafety is not a concern in the development of a new generation of ASF vaccines, such as subunit, mRNA, or viral vector vaccines. To date, these vaccines have not been used because of their lack of protective effect, even though animals can be induced to produce a strong immune response by them [14,15]. 

The assessment of the immunoprotective efficacy of a vaccine is an important part of the vaccine development process, and the current common method is usually the intramuscular injection of a highly virulent ASFV strain. However, the challenge of intramuscular injection does not fully mimic natural infection because it bypasses innate defence mechanisms, such as the interaction of the virus with the mucosal surfaces of the mouth and upper respiratory tract. In addition, during our vaccine development, it was found that some candidate vaccine strains, such as ASFV SY18ΔMGF/ΔCD2v with the MGFs and CD2v genes deleted, could induce sufficient protection in pigs against oral challenge with the parental highly virulent strain, but not against intramuscular challenge with the same strain [16,17]. The method of challenge affects the outcome of the assessment of immune protection against the ASFV vaccine, and an intramuscular challenge alone is not comprehensive enough.

In this study, using established and recommended challenge infection models, we performed an infection trial in pigs using different administration routes and different infectious doses of a highly lethal ASFV strain and observed disease progression, thereby exploring the potential of intraoral (IO) infection and intranasal (IN) infection as alternatives to intramuscular (IM) infection as challenge models for ASFV vaccine efficacy testing trials. 

## 2. Materials and Methods

### 2.1. Virus 

The ASFV SY18 strain (GenBank accession No. MH766894.2) is a field isolate from the first ASF outbreak in China in 2018. It is highly homologous to the 100% lethal strain (Georgia 2007/1) and has been used to develop a live vaccine candidate at the Epidemiology Laboratory of Changchun Veterinary Research Institute, Chinese Academy of Agriculture [16,18]. The fourth-generation primary porcine alveolar macrophage (PAMs) virus-containing cell culture, stored at −80°C in a biosecurity level 3 lab, was used in this animal experiment. 

The virus titres were determined using the fluorescein isothiocyanate (FITC)-labelled ASFV p30 monoclonal antibody direct fluorescence method (developed in our laboratory using the hybridoma technique, reference from NCBI’s GeneID:59226957) with 50% tissue culture infective dose per millilitre (TCID_50_/mL), as previously described [18,19].

### 2.2. Pigs 

In each infection trial with different challenge routes, landrace pigs (30–45-day-old, weighing 10–15 kg) were purchased from a local farm in Changchun, Jilin, with high biosecurity and hygiene standards. To ensure that the animals were not infected with ASFV, porcine reproductive and respiratory syndrome (PRRSV), classical swine fever virus (CSFV), porcine circovirus 1/2 (PCV1/2), porcine parvovirus (PPV) and pseudorabies virus (PRV), polymerase chain reaction (PCR) or quantitative PCR (qPCR) were performed according to the assays that have been described [18]. The INGEZIM PPA COMPAC enzyme-linked immunosorbent assay (ELISA) kit (Ingenasa, Madrid, Spain) was used to determine the absence of ASFV p72 antibodies in the sera of pigs according to the manufacturer’s instructions. Pigs were randomly and equally divided into different experimental groups (Table 1), bred in independent beds, and continuously observed for 7 days. 

### 2.3. Animal Experiments 

The animal experiments were conducted under animal biosecurity level 3 (ABSL-3) conditions and approved by the Animal Welfare and Ethics Committee of the Changchun Veterinary Research Institute, Chinese Academy of Agriculture (review ID: IACUC of CAS-12-2021-011, approved on 1 December 2021).

The details of three independent infection groups are shown in Table 1. The first group was a control group named IM200, in which three pigs were injected intramuscularly with 200 TCID_50_ doses of ASFV SY18 (in 2 mL of cell culture medium) in the gluteal region, (the minimum intramuscular dose of the published ASFV SY18 challenge experiment (10^2.5^ TCID_50_ and 10^1.0^ TCID_50_) [16,20]). In the second group, nine pigs were equally divided into three subgroups (n = 3 each) and intranasally infected with ASFV SY18 at doses of 40, 200, and 1000 TCID_50_ (in 2ml of cell culture medium). The third group involved IO infection, in which pigs in four subgroups (n = 3 each) were inoculated intraorally with ASFV SY18 at doses of 40, 200, 1000, and 5000 TCID_50_ (in 2 mL of cell culture medium). 

The specific infection method for group IN and group IO was as follows: the handler holds the piglet close to the body, wraps one hand firmly around the piglet’s back, supports the abdomen, and cradles the piglet’s neck and head with the other hand. Another handler uses a combination of a nozzle and syringe to administer the virus, as shown in Appendix A (see Appendix A). Specifically, the nozzle is into the mouth (IO route) or the nostril (IN route) and slowly pushed into the syringe so that the virus-containing liquid (2 mL) enters the oral or nasal cavity in the form of an aerosol.

### 2.4. Observation of Clinical Presentations

The clinical scoring criteria were developed based on the ASF clinical scoring grid, as previously described by King et al. (see Appendix A) [21]. All pigs were observed daily for 28 days, and clinical signs (rectal temperature, appearance, appetite, respiratory performance, behaviour, faecal, and skin changes) were measured. Pigs with clinical scores greater than or equal to 14 were euthanised using pentobarbital sodium. Pigs were also euthanised if they survived to day 28 after injection.

### 2.5. Sample Collection

Swab samples were collected at 10 am from the injection day with ASFV SY18. Briefly, oral swab samples were collected by scraping the lining of the mouth with the swab, collecting saliva, and storing it in 1 mL of saline. Similarly, faeces were collected as rectal swab samples.

Blood samples were collected and stored in EDTA tubes after the pigs had been euthanised. This was done to avoid frequent live blood collection causing internal bleeding and stress reactions in the pigs, which could affect the disease process.

Tissue (heart, liver, spleen, lung, kidney, thymus, submandibular lymph nodes, inguinal lymph nodes, bone marrow and intestine) were also collected at the post-mortem session after the euthanasia of the piglets. Two samples (0.5 g each) were collected from each tissue, one fixed in paraformaldehyde for pathological analysis and the other homogenised with 0.5 mL of saline and the supernatant was taken for viral load analysis.

DNA from all samples was extracted using the Axygen Body Fluid/Viral DNA Extraction Kit (Corning, Wujiang, China) and stored at −80 °C.

### 2.6. ASFV Viral Load Analysis

All swabs and tissue samples were analysed for ASFV nucleic acids, according to the method recommended by The World Organization for Animal Health (WOAH) (Chapter 3.9.1 of the Manual of Diagnostic Tests and Vaccines for Terrestrial Animals 2022). Briefly, ASFV genomic DNA was quantified in samples using qPCR, targeting the ASFV *B646L* (p72) gene. Primers were as described by King et al., with the forward primer 5′-CTGCTCATGGTATCAATCTTATCGA-3′, the reverse primer 5′-GATACCACAAGATCAGCCGT-3′, and the Taqman probe FAM-5′-CCACGGGAGGAATACCAACCCAGTG-3′-TAMRA [22]. The extracted DNA was used for qPCR, which was performed using LightCycler^®^ 480 (Roche, Basel, Switzerland) and Premix Ex Taq™ (Probe qPCR) reagents (Takara, Dalian, China). The reaction system was based on Takara’s product instructions and the amplification conditions were 95 °C pre-warm for 30 s, 40 cycles of 95 °C for 5 s, and 60 °C for 30 s. 

ASFV *B646L* (p72) gene copy number was calculated using a standard curve, generated to detect the p72 plasmid (constructed in our laboratory). The equation was *y* = −3.3236*x* + 49.224, where *y* is the cycle threshold (C_t_) value, and the unit of *x* is log10 copies/mL. The amplification efficiency (*e*) was 99.2%, and the correlation coefficient (R^2^) was 0.9968. The minimum detected concentration was 10^3^ copies/mL.

### 2.7. Statistical Analysis

Based on the data collected in this experiment, figures were plotted using GraphPad Prism 9.4.1 software (https://www.graphpad.com, accessed on 3 November 2022), and selected data (disease process and virological parameters) were analysed using the software’s built-in one-way ANOVA and Tukey’s post-hoc test. 

## 3. Results

### 3.1. Disease Progression and Clinical Signs

The twenty-four pigs were divided into three groups according to the route of infection, and eight subgroups according to the dose of infection (see Table 1 for specific group information). It was observed that the IM control group began to develop fever on day 5 (n = 2) and day 8 (n = 1) after inoculation, and all showed typical symptoms of ASF. Individuals in the three subgroups in the IN group also began to develop fever symptoms on day 5 and continued to develop typical ASF symptoms, with similar incubation periods in each subgroup. However, in the IO group, the fever began to appear around the 15th day in the low-dose subgroup (IO40) and around the 11th day in the high-dose subgroup (IO5000). The incubation period of the low-dose subgroup was longer than that of the high-dose subgroup, but there was no significant difference between the subgroups. In general, the incubation period of the IN group (5.3 ± 1.1 days) was similar to that of the IM group (5.0 ± 1.7 days), both of which were significantly shorter than that of the IO group (13.3 ± 2.3 days), as shown in Figure 1 and Table 2 (Pyrexia Days to onset).

Although the incubation period of the IO group was different from that of the other two infection modes, it was observed that the symptoms of individuals in the three infection mode groups after the onset of fever were typical symptoms of ASF. No significant differences occurred during the 28-day observation period (Table 3), with all pigs first showing signs of high fever (≥40.5 °C), and 2–3 days later, a loss of appetite and reluctance to stand were observed. Other clinical signs observed in more than 1/3 of all individuals were cough, enlarged joints, skin purpura, and diarrhoea. Of these, coughing was observed in 6/9 of the pigs infected via the IN route. This was significantly higher than in the IO and IM groups (1/3 and 4/12, respectively). Ocular discharge and bloody diarrhoea occurred less frequently and were only observed in 6/24 pigs in the overall animal experiment. 

High fever (≥40.5 °C) is the earliest sign of disease that manifests itself in pigs after infection with highly virulent ASFV [2]. In this study, we calculated the time from the onset of high fever until the animals were euthanised as the duration of disease after ASF onset. The duration of disease generally increased in the IN and IO groups (4.9 (±0.8) and (4.6 ± 0.8) days, respectively) compared to 4.0 (±1.0) days in the control group (IM200), but the difference was not significant (Table 2, Duration).

For the overall survival time after challenge, there was no significant difference between the IN group and the IM group, both were about 10 days after infection, while it was significantly longer in the IO group (17.9 (±2.3) days). Within each subgroup, i.e., at the same route of inoculation and at the same dose of inoculation, the difference in survival time was less than 4 days, with only individuals within the IO40 subgroup showing a more significant difference in survival time (>6 days) (Table 2, Total days survival).

### 3.2. Virus Shedding via Oral and Rectal Routes

To detect the shedding of ASFV via the oral and rectal routes, oral and rectal swabs were collected daily and the viral load was evaluated using quantitative qPCR. The test results showed that viral nucleic acid appeared in oral and rectal swabs on day 5 in the IM control group (Figure 2). There was no significant difference between rectal and oral virus shedding time. The oral virus shedding time of the IN group was about 6.7 (±1.3) days and the rectal virus shedding time was about 6.3 (±1.2) days, both of which were not significantly different from the IM control group. The oral and rectal virus shedding time was also not significantly different between different dose subgroups in the IN group. However, the oral and rectal virus shedding time of the IO group was about 14.4 days and 13.9 days, respectively, both of which were significantly later than the IM control group and the IN group. This result is consistent with the difference between the infection mode groups in the incubation period. In addition, among the IO dose subgroups, the virus shedding time of the highest dose subgroup (5000 TCID_50_) was slightly earlier than that of the other three dose subgroups. Still, there was no significant difference among all groups. The oral and rectal virus shedding time between individuals in the lowest dose subgroup (40 TCID_50_) was significantly different. For IO infection, although the low dose group could also cause typical ASF symptoms, the consistency between individuals in the subgroup was poor.

In general, the virus shedding time of the IO group was significantly later than that of the IM and IN injection groups, and there was no significant difference between the latter two groups. However, viral load in oral swabs and rectal swabs increased daily to 10^6^ copies/mL after virus shedding began in all individuals, and there was no significant difference in the maximum viral load in swabs among all subgroups (Table 2, Maximum titer). In addition, viral load was higher and detected first in rectal swabs when compared to oral swabs; however, still, no significant difference was observed.

### 3.3. Viral Load in Tissues and Blood

Using fluorescence qPCR, the ASFV *B646L* (p72) gene was quantitatively detected in the heart, liver, spleen, lungs, kidneys, thymus, mandibular lymph nodes, inguinal lymph nodes, intestinal tract, blood, and bone marrow of all animals after death. The results showed that viral nucleic acids were present in all detected tissues and organs at the time of animal death (Figure 3). The viral load in the blood, bone marrow, spleen, liver, and lungs were generally higher than that in the heart, kidney, thymus, lymph nodes, and intestines for all groups, but the differences were not significant. In addition, there was no significant difference in viral gene content between the groups of different infection routes and doses. Namely, the infection route and dose did not significantly affect the distribution of the virus in the organs of major tissues at death.

### 3.4. Post-Mortem Lesions

A necropsy was performed after death or euthanasia for all individuals, and typical ASF necropsy changes were generally observed. As shown in Table 3, 100% of individuals in all infection mode groups and all dose subgroups showed, to varying degrees, thickened, dark, and friable spleens. Although, to different degrees, enlarged spleens also appeared in most individuals, and the frequency of occurrence was not significantly different or regular among all subgroups.

In addition, the thymus and kidneys in 100% of pigs; the liver, lungs, submandibular, and inguinal lymph nodes in over 80% of pigs; and the heart in 60% of pigs had varying degrees of pathology. Among them, the lesions of the thymus were mainly haemorrhagic, the kidney lesions were mainly characterised by petechiae, the liver was mainly characterised by enlargement and friable, and the lung was mainly characterised by interstitial pneumonia and interlobular oedema. The lymph nodes were characterised by enlargement and haemorrhage, and the cardiac lesions were characterised by epicardial haemorrhage. Hydropericardium was found in a few individuals in the IN and IO groups but not in the IM control group. In general, there was no significant difference between the infection mode groups and the dose subgroups in the various tissues and organs at the time of death, and this was consistent with the reported typical acute ASF infection (Figure 4) [2].

### 3.5. Pathological Changes

Consistent with the necropsy results, the types of pathological changes in major tissues and organs were basically the same among all groups, except that there were differences in the degree of pathological changes among individuals. Still, such differences had no obvious correlation with challenge methods and doses. The typical pathological changes observed included normal cardiomyocytes with occasional hyaline proliferation of fibrous tissue, common inflammatory cell infiltration in the liver with congestion and enlargement of the hepatic sinuses, necrotic cells in the spleen with broken nuclei, reduced lymphocytes, partial alveolar atrophy with inflammatory cell infiltration in the lungs, oedematous dilatation or necrosis of the renal tubules in the atrophic kidneys, erosion and detachment of intestinal epithelial cells, necrotic degeneration of cells in the thymus, inguinal lymph nodes, and submandibular lymph nodes (Figure 5).

## 4. Discussion

ASF is an acute infectious disease that has a 100% mortality rate among pigs [1]. Since 2007, the spread of genotype II ASFV in Eastern Europe, the Caucasus, East Asia, Southeast Asia, and the Caribbean has created an urgent need for an ASFV vaccine [23,24,25,26,27,28]. To evaluate and validate the effectiveness of different ASF vaccines, it is necessary to perform a challenge test with virulent ASFV. IM infection has been used in many in vivo ASFV studies because it produces a highly reproducible clinical disease [29,30]. However, the evaluation of vaccine protection using only IM infection is incomplete, as pigs are resistant to oral challenge but not to intramuscular challenge after vaccination with some ASFV vaccine strains [16,17]. Therefore, the development of alternative routes of infection is necessary to assess the immunoprotective effects of vaccines.

In this study, to establish a challenge model for ASFV intraoral and intranasal infections, SY18 was selected as the challenge strain. The ASFV SY18 strain, a genotype II virulent strain isolated from dead pigs in Shenyang, the site of the first ASF outbreak in China, is highly homologous to the classic challenge strain, ASFV Georgia 2007/1 (GenBank: FR682468.2). IM infection of pigs with ASFV SY18 at 10^2.0^ TCID_50_ resulted in 100% mortality [16,18,20]. Our laboratory infected pigs with ASFV SY18 via IM at doses ranging from 1 TCID_50_ to 10^4.0^ TCID_50_ during the course of developing a gene deletion vaccine, with no significant differences in disease duration (see Appendix A). We infected pigs with SY18 via the IN and IO route and compared disease development with that of IM infections. The results showed that IN or IO SY18 infection of pigs causes the same acute ASF symptoms as that observed in pigs infected via the IM route, ultimately leading to death. All pigs developed fever and loss of appetite following infection. In addition, typical lesions of acute ASF, such as thickened, darkened, and enlarged spleens, were observed during necropsy. Viral shedding was similar, with viral nucleic acids detected in secretions collected from the mouth and rectum when animals passed the incubation period following ASFV infection and began to show clinical signs (fever, anal temperature ≥40.5 °C). Viral shedding then increased as the disease progressed, reaching a peak (10^5.5^–10^6.0^ copies/mL) and remaining there until the animal dies. For most individuals in this study, viral nucleic acid was detected slightly earlier in rectal swabs compared to oral swabs, and the virus content was also slightly higher, with no significant difference. This result is inconsistent with previous studies, and it is speculated that the reason may be that the amount of material in the rectal swab could be more than that of the oral swab. These differences were minimal, not statistically significant, and did not affect our judgement of the characteristics of the viral infection. 

During necropsy, identical pathological changes were observed in all inner organs. The viral loads measured in different groups and animals were similar, with higher viral loads in the blood, liver, spleen, lungs, and bone marrow, which may be related to the abundance of mononuclear macrophages, such as PAMs and bone marrow macrophages (BMDM), in these tissues and organs. IM, IN, and IO modes of infection caused acute ASF symptoms that led to death. IN and IO infection could be used as IM alternatives as these results were consistent with the findings of previous studies. 

The minimum ASF dose that causes 100% mortality has not been identified. In addition, the criteria for challenging viral doses are yet to be established, particularly in the context of vaccine development. In 2013, Howey et al. infected pigs with the ASFV Malawi strain at a dose of 10^2.0^ HAD_50_. IO infection was unsuccessful, and only one of two pigs were infected by IN [31]. In 2019, Niederwerder et al. infected pigs with the ASFV Georgia 2007/1 strain using infected drinking water and a minimum infectious dose of 1 TCID_50_ was used [32]. In our study, IN and IO infections using 40 TCID_50_ resulted in severe clinical signs being observed in pigs within 11 and 23 days, respectively. In addition, disease progression of IN infection within the dose range 40–1000 TCID_50_ overlapped significantly, which is consistent with the results of Howey et al., who challenged pigs via the IN route using a highly virulent strain at doses of 10^3.0^ TCID_50_-10^6.0^ TCID_50_ [31]. This suggests that disease progression following infection via the IN route is less influenced by dose if a higher than minimal infectious dose is used. Many factors contribute to results, such as pigs breed, age, and the environment in which they are kept. However, since the same pigs were used in this study and the only variable was mode of infection, we believe that the mode of infection is a critical factor in disease development. IN or IO infections are performed using a nozzle that produces water mist and allows the virus-containing fluid to enter the body cavity in an atomised form, which ensures effective and consistent infection. 

The main difference observed between IN, IO, and IM infections was the incubation period, that is, the number of days between infection and symptomatic presentation. The incubation period for IM, IN, and IO infection were IM 5.0 ± 1.7 days, IN 5.3±1.1 days, and IO 13.3 ± 2.3 days, respectively. In IM infection, the virus enters the bloodstream directly and quickly contacts the monocytes and macrophages, which are the target cells of ASFV, leading to a rapid disease process. In contrast, the IO infection process is close to natural infection, with the virus first coming into contact with the mucosa and inducing an innate immune response, followed by infection of the lymph nodes and then monocytes and macrophages, exhibiting a more extended latency period [31,33,34]. However, in this study, IN infection also resulted in rapid death. We speculate that after IN administration, the virus solution, which the nozzle presses into an aerosol, enters the lungs directly with the animal’s breathing. If this occurs, the virus bypasses the innate immune response in the mucosa and directly infects the PAMs, where it replicates in large numbers, accelerating disease progression. 

In summary, infection of pigs with a specific dose (≥40 TCID_50_) of the genotype II highly virulent strain ASFV SY18 via the IN, IO, and IM routes all stably caused typical acute ASF symptoms with similar viral shedding, viral load in tissues, and pathological changes. Therefore, IO or IN infection challenges can be used as a complementary assessment method for ASF vaccines to allow a more comprehensive evaluation of inactivated and live vector vaccine candidates. Although, it must be admitted that an ASF vaccine could prevent IM infection challenge by a virulent ASFV strain, indicating that the vaccine produced a stronger immune response; inhibiting early, direct, and rapid replication of the challenged virus.

## 5. Conclusions

This study evaluated a challenge model for ASFV SY18 infection using oral and nasal sprays. Pigs infected intranasally with 40–1000 TCID_50_ and orally with 40–5000 TCID_50_ exhibited consistent acute clinical forms of ASF infection that did not differ significantly from that observed in pigs infected intramuscularly. Clinical signs, viral shedding, and histopathological changes were comparable across all groups. However, the incubation period for intraoral infection was 13.3 ± 2.3 days, which was significantly longer than the 5.0 ± 1.7 and 5.3 ± 1.1 days as observed in pigs infected intramuscularly and intranasally, respectively.

This study showed that intraoral and intranasal infections can be used as alternatives to intramuscular infections in ASFV challenge trials.

## Figures and Tables

**Figure 1 viruses-15-00858-f001:**
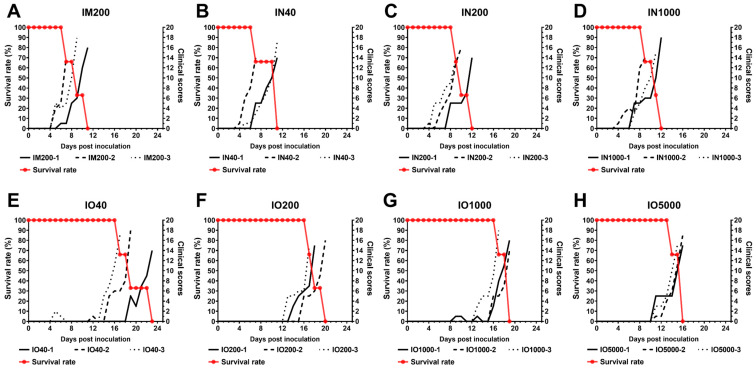
Survival rates and clinical scores in each group. (**A**) Inoculation of ASFV SY18 at 200 TCID_50_ via the IM route; (**B**–**D**) Inoculation of ASFV SY18 at 40, 200, and 10,000 TCID_50_ via the IN route; (**E**–**H**) Inoculation of ASFV SY18 at 40, 200, 1000, and 5000 TCID_50_ via the IO route. The line indicating the clinical score for each pig is shown on the right Y-axis. The line indicating the group’s survival rate is on the left Y-axis.

**Figure 2 viruses-15-00858-f002:**
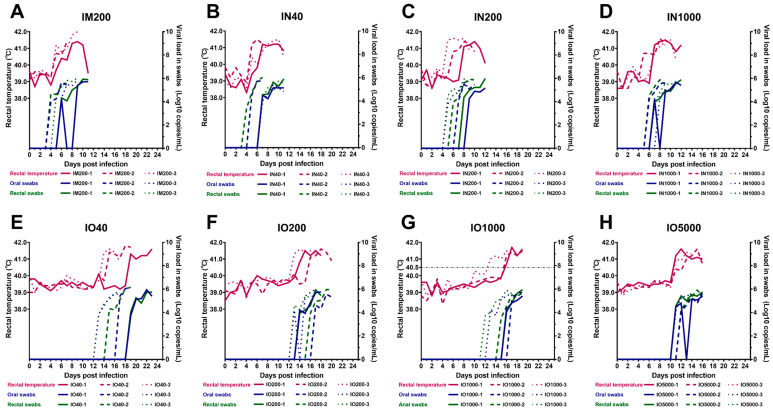
Results of rectal temperature and viral load in swabs for each group. (**A**) Inoculation of ASFV SY18 at 200 TCID_50_ via the IM route; (**B**–**D**) Inoculation of ASFV SY18 at 40, 200, and 10,000 TCID_50_ via the IN route; (**E**–**H**) Inoculation of ASFV SY18 at 40, 200, 1000, and 5000 TCID_50_ via the IO route. The lines indicating rectal temperature (°C) are on the left Y-axis. The viral load in the swab (Log10 copies/mL) is indicated on the right Y-axis.

**Figure 3 viruses-15-00858-f003:**
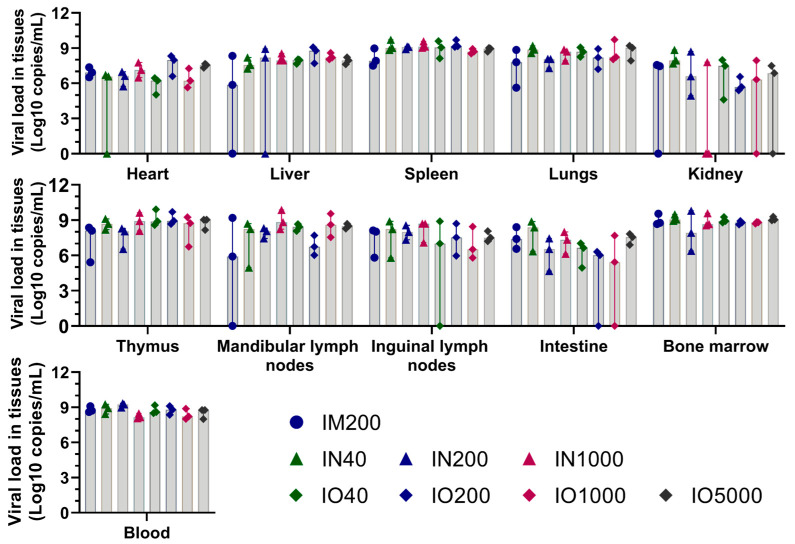
Viral load in tissues and blood. ASFV viral load in tissues (heart, liver, spleen, lungs, kidneys, thymus, mandibular lymph nodes, inguinal lymph nodes, intestines, bone marrow) and blood from pigs in the IM group (IM200), IN group (IN40, IN200, IN1000), and IO groups (IO40, IO200, IO1000 and IO5000).

**Figure 4 viruses-15-00858-f004:**
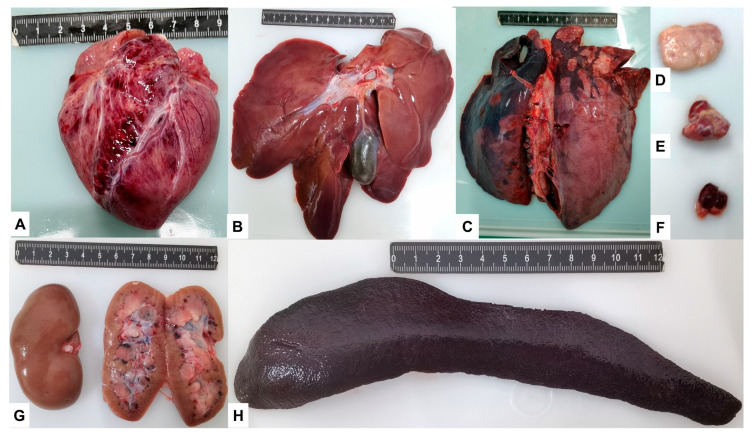
Gross lesions at autopsy. (**A**) Heart with myocardial haemorrhage from pig No. IN200-2; (**B**) Enlarged and hardened liver from pig No. IM200-1; (**C**) Necrotic lung from pig No. IO40-3; (**D**–**F**) Enlarged and necrotic inguinal lymph nodes, submandibular lymph nodes, and thymus from pig No. IO5000-1; (**G**) Kidney with bleeding spots from pig No. IM200-2; (**H**) Severely enlarged spleen from pig No. IO200-2.

**Figure 5 viruses-15-00858-f005:**
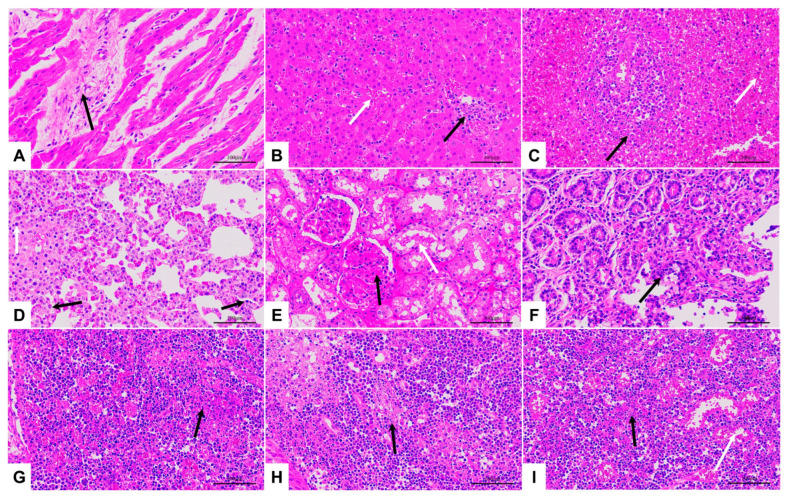
Histopathological section (H and E staining, 400×) (**A**) Heart from pig No. IM200-2, fibrous tissue hyperplasia visible in areas (black arrow) of the myocardial interstitium; (**B**) Liver from pig No. IN40-2, congested and enlarged hepatic sinusoids (white arrows) with inflammatory cell infiltration (black arrows); (**C**) Spleen from pig No. IO200-2, atrophy of splenic nodules, massive cell necrosis, broken nuclei (black arrows), and reduced lymphocytes (white arrows); (**D**) Lung from pig No. IO40-3, partial alveolar atrophy with inflammatory cell infiltration (black arrow) and cells with fragmented nuclei (black arrow); (**E**) A kidney from pig No. IO100-2, tubular epithelial cells with oedema, loose vacuolation of the cytoplasm and dilated lumen (white arrow), glomerular amyloidosis, thickened basement membrane, and poorly defined capillary collaterals (black arrow); (**F**) Intestine from pig No. IM200-1, epithelial cells of the mucosal layer are eroded and detached, the lamina propria is exposed, and limited infiltration of inflammatory cells is visible (black arrow); (**G**) Thymus from pig No. IO200-3, necrotic degeneration of cells with fragmented nuclei (black arrow); (**H**) Inguinal lymph node from pig No. IO1000-2, with necrotic degeneration of lymphocytes, and fragmentation of nuclei (black arrow); (**I**) Submandibular lymph node from pig No. IO200-3, necrotic degeneration of lymphocytes with fragmented nuclei (black arrow), and extensive vasodilation and congestion (white arrow).

**Table 1 viruses-15-00858-t001:** Experimental groups.

Groups	Subgroups	Routes of Injection	Dose	Pigs (n)	No.
IM	IM200	intramuscular	200 TCID_50_	3	IM200-1 IM200-2 IM200-3
IN	IN40	intranasal	40 TCID_50_	3	IN40-1 IN40-2 IN40-3
	IN200	intranasal	200 TCID_50_	3	IN200-1 IN200-2 IN200-3
	IN1000	intranasal	1000 TCID_50_	3	IN1000-1 IN1000-2 IN1000-3
IO	IO40	intraoral	40 TCID_50_	3	IO40-1 IO40-2 IO40-3
	IO200	intraoral	200 TCID_50_	3	IO200-1 IO200-2 IO200-3
	IO1000	intraoral	1000 TCID_50_	3	IO1000-1 IO1000-2 IO1000-3
	IO5000	intraoral	5000 TCID_50_	3	IO5000-1 IO5000-2 IO5000-3

**Table 2 viruses-15-00858-t002:** Disease processes and virological parameters in pigs infected with ASFV SY18 by different routes and doses.

Group		No. Infected/Total	Disease Process			Viral Load in the Shedding			
			Pyrexia Days to Onset (±SD) ^a^	Duration (±SD)	Total Days Survival (±SD)	Oral Shedding		Rectal Shedding	
						Days to the Onset (±SD)	Maximum Titer (±SD) ^b^	Days to the Onset (±SD)	Maximum Titer (±SD) ^b^
IM	IM200	3/3	5.0(±1.7)	4.0(±1.0)	9.0(±2.0)	5.3 (±1.2)	5.8 (±0.1)	5.0 (±1.0)	5.8 (±0.2)
IN	Overall	9/9	5.3(±1.1) ^d^	4.9(±0.8) ^d^	10.2(±1.6) ^d^	6.7 (±1.3) ^d^	5.5 (±0.2) ^d^	6.3 (±1.2) ^d^	5.9 (±0.1) ^d^
	IN40	3/3	5.3(±1.2)	4.3(±1.2)	9.7(±2.3)	6.3 (±1.2)	5.4 (±0.3)	6.0 (±1.7)	5.9 (±0.2)
	IN200	3/3	5.3(±1.7)	5.0(±0.0)	10.3(±1.5)	7.0 (±2.0)	5.4 (±0.2)	6.3 (±15)	6.0 (±0)
	IN1000	3/3	5.3(±1.2)	5.3(±0.6)	11.7(±1.5)	7.0 (±1.0)	5.6 (±0.2)	6.7 (±0.6)	5.9 (±0.1)
IO	Overall	12/12	13.3(±2.3) ^c, d^	4.6(±0.8) ^d^	17.9(±2.3) ^c, d^	14.4 (±2.7) ^c, d^	5.7 (±0.2) ^d^	13.9 (±2.4) ^c, d^	5.8 (±0.2) ^c, d^
	IO40	3/3	15.0(±2.6) ^c^	4.7(±0.6)	19.7(±3.1) ^c^	16.3 (±3.1) ^c^	5.9 (±0.3)	15.7 (±3.1) ^c^	5.8 (±0.3)
	IO200	3/3	13.3(±1.5) ^c^	5.0(±0.0)	18.3(±1.5) ^c^	14.7 (±2.1) ^c^	5.8 (±0.2)	14.3 (±1.5) ^c^	5.8 (±0.3)
	IO1000	3/3	14.0(±1.7) ^c^	4.3(±0.6)	18.3(±1.2) ^c^	15.3 (±2.1) ^c^	5.6 (±0.2)	14.3 (±2.1) ^c^	5.8 (±0.1)
	IO5000	3/3	11.0(±1.7) ^c^	4.3(±1.5)	15.3(±1.2) ^c^	11.3 (±0.6) ^c^	5.6 (±0.2)	11.3 (±0.6) ^c^	5.8 (±0.1)

^a^ Rectal temperature ≥ 40.5 °C. ^b^ Log10 copies/mL. ^c^ The value of the group is significantly different (*p >* 0.05) from the control group (Group IM200). ^d^ The values between the subgroup of the same challenge routes are not significantly different (*p* < 0.05) from each other.

**Table 3 viruses-15-00858-t003:** Summary of clinical signs and gross lesions in pigs inoculated with different doses of ASFV SY18 at different modes of infection.

Clinical Signs and Gross Lesions			IM	IN				IO				
			IM200	Overall	IN40	IN200	IN1000	Overall	IO40	IO200	IO1000	IO5000
Clinical Signs	Fever (≥40.5 °C)		3/3	9/9	3/3	3/3	3/3	12/12	3/3	3/3	3/3	3/3
	Lost appetite		3/3	9/9	3/3	3/3	3/3	12/12	3/3	3/3	3/3	3/3
	Reluctance to stand		3/3	9/9	3/3	3/3	3/3	12/12	3/3	3/3	3/3	3/3
	Wheezing/ coughing		1/3	6/9	2/3	1/3	3/3	4/12	1/3	0/3	2/3	1/3
	Ocular discharge		1/3	2/9	0/3	1/3	1/3	3/12	2/3	0/3	1/3	0/3
	Enlarged joints		2/3	5/9	1/3	2/3	2/3	6/12	2/3	2/3	1/3	1/3
	Skin Purpura		1/3	4/9	2/3	1/3	1/3	6/12	2/3	1/3	2/3	1/3
	Diarrhoea		2/3	2/9	1/3	0/3	1/3	5/12	2/3	1/3	1/3	1/3
	Bloody diarrhoea		0/3	2/9	1/3	0/3	1/3	4/12	1/3	0/3	0/3	1/3
Gross lesions	Heart	Epicardial haemorrhage	2/3	5/9	2/3	1/3	2/3	7/12	2/3	2/3	3/3	1/3
		Hydropericardium	0/3	2/9	0/3	1/3	1/3	2/12	1/3	0/3	1/3	0/3
	Liver	Enlargement	2/3	7/9	2/3	3/3	2/3	10/12	3/3	2/3	3/3	2/3
		Friable	3/3	8/9	3/3	3/3	2/3	12/12	3/3	3/3	3/3	3/3
	Spleen	Thickened	3/3	9/9	3/3	3/3	3/3	12/12	3/3	3/3	3/3	3/3
		Dark	3/3	9/9	3/3	3/3	3/3	12/12	3/3	3/3	3/3	3/3
		Enlarged	3/3	7/9	2/3	3/3	2/3	9/12	3/3	2/3	2/3	2/3
		Friable	3/3	9/9	3/3	3/3	3/3	12/12	3/3	3/3	3/3	3/3
	Lungs	Interstitial pneumonia	2/3	7/9	2/3	2/3	3/3	9/12	2/3	2/3	2/3	3/3
		Interlobular edema	3/3	9/9	3/3	3/3	3/3	11/12	3/3	3/3	2/3	3/3
	Kidney	Petechiae	3/3	9/9	3/3	3/3	3/3	12/12	3/3	3/3	3/3	3/3
	Thymus	Haemorrhage	3/3	9/9	3/3	3/3	3/3	12/12	3/3	3/3	3/3	3/3
	Submaxillary lymph nodes	Enlargement	2/3	8/9	3/3	2/3	3/3	12/12	3/3	3/3	3/3	3/3
		Haemorrhage	3/3	7/9	2/3	3/3	2/3	11/12	3/3	3/3	2/3	3/3
	Inguinal lymph nodes	Enlargement	3/3	8/9	3/3	3/3	2/3	10/12	2/3	3/3	2/3	3/3
		Haemorrhage	1/3	4/9	1/3	2/3	1/3	6/12	1/3	2/3	2/3	1/3

## Data Availability

Not applicable.

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
