# Peer review of "Comparison of Genotype II African Swine Fever Virus Strain SY18 Challenge Models"

_viruses, 2023, doi:10.3390/v15040858_

Round 1

Reviewer 1 Report

Article “Comparison of genotype II African swine fever virus strain SY18 challenge models for the evaluation of newly developed vaccine candidates”

The manuscript presents a well-designed experiment where alternative routes to IM of ASF challenge were evaluated. The authors were using SY18 highly pathogenic Genotype -II virus under lethal challenge 100 using different routes of administration: intramuscular, intranasal and intraoral. For the IN and IO different doses were used. The authors did not find many differences among the routes of administration or doses apart from the onset of clinical symptoms. The overall manuscript is well presented but there is room for improvement.

Minor comments:

-        Line 46. African swine fever virus doesn’t have 234 open reading frames, the different isolates have between 151 and 167 (Alejo et al., 2018).

-        Line 49. The authors are categorizing the outcome of infection in two different ways and this can confuse the reader. Please, stick to “severe, moderate or mild” or “peracute, acute, subacute, chronic or subclinical”.

-        Line 53. No vaccine nor treatment are available.

-        Line 61. The authors should clarify why the adaptation to the host in pig farms that have illegally used the ASFV-G-DI177L is a biosafety concern.

-        Line 75-76, “and observed disease progression” is repeated.

-        Line 110. The results of the experiment are not presented in table 1, what is presented is the experimental groups details. Please, correct.

-        Material and methods section 2.3 Animal experiments. The rationale behind the doses used is not properly explained. Please, correct.

-        Material and methods section 2.5. Sample collection. It is not stated that tissue samples were stored or DNA isolation. Please, correct.

-        Material and methods section 2.6. ASFV viral load analysis. This section needs revision and be better structured. The authors are stating they were following King’s instructions but later on they changed the amplification cycles for the Takara’s ones.

-        Line 181, please correct the following “developed fevers at 5, 5 and 5 dpi”.

-        Line 200. I think the authors wanted to say “the difference between the groups was not significant”. It’s only significant for IO but not IN.

-        Figure 1. The headline of the different panels is the same it should indicate the name of the groups. Please, correct.

-        Figure 2. The title of the panels is not correct, and the legend has mistakes. Please, correct.

-        Line 231. The authors state “ASFV genes were quantitatively detected”, which genes?

-        Figure 3. Please, improve the figure to know where one group starts and the other finish. It will be better to group by tissue and not by animals for a better comparison.

-        Line 296-298. “it was found that some recombinant virus candidate strains could not resist intramuscular challenge after immunisation, but could resist oral challenge (data not shown).” The viruses are not resisting the challenge, the animals immunized with the virus do.

-        Line 313. “such as necrosis of the lungs” is not an ASF typical lesion. Please, reference your statement.

-        Line 318 to 320. Please, rephrase the stamen “There was no significant difference in ASFV viral load between oral and anal swabs, across all groups, however, earlier detoxification and higher viral gene content was observed in anal swabs”.

-        Line 334-335. “resulted in the total near death of pigs”. Please, correct this sentence.

-        Line 351-353. Please reference this statemenet “n contrast, IO infection mimics natural infection, that is, the virus enters the mouth, passes through the oral mucosa and induces an innate immune response, as indicated by the longer incubation period, slower disease progression and eventual death”

-        Line 376. The authors should remove this statement, it is not supported by the data.

Main comments:

-        Line 65 to 67. Subunit vaccine induce strong immune response, sometimes not protective, but the immune response is strong. The subunit vaccines are not being used because they don’t protect, but they are immunogenic. Please, correct and reference your statements.  

-        Line 67 to 70. The statement that subunit vaccines are not efficacious against lethal doses of ASFV because they only contain portions of the virus is incorrect and need revision or a reference to support the statement. “Based on our findings and conversations with other researchers, the dose of the challenge virus used for the gene-deleted ASF vaccine is usually 102.0 50% hemadsorbing doses (HAD50) to 104.0 HAD50, and this dose is too high for the efficacy evaluation of a subunit vaccine, which contains only some components of the ASFV.”

-        Line 70 to 72: The statement “In addition, challenge methods significantly impact the induced immune response, and subunit vaccines usually induce an immune response that is resistant to oral rather than intramuscular challenges.” Needs revision or a reference to support the statement.

-        The introduction should focus more on the LAV and the potential of the challenge models to evaluate their efficacy and cross-protection than on the subunit vaccines that are not the potential immediate users of the presented challenge models.

-        From line 197 to 204. The authors need to review this section since there are statements only applicable to one group of animals and it’s been generalized to all groups.

-        Section 3.2. This section is very confusing and needs extensive revision. It contains statements that contradict each other and it’s not well structured.

-        Section 3.4. The authors should expand on the macroscopic lesions and differences between groups, even if differences are slight.

-        Section 3.5. The authors should focus on the differences between groups or doses in greater detail. The information provided is not very informative of this specific experiment.

-        These two statemenets contradict each other “IN and IO infection could be used as IM alternatives, these results were consistent with the findings of previous studies. However, in our study, a different effective infectious dose was observed.”

-        Based on your results this statement is not correct “In our study, IN and IO infection using 40 TCID50 resulted in the total near death of pigs within 11 and 23 days, respectively. In addition, disease progression of IN infection within the dose range 40-1000 TCID50 overlapped significantly, this suggests that 40 TCID50 is not the minimum infectious dose for IN infection.”

-        Line 339 -344. This statement “However, since the same pigs were used in this study and the only variable was mode of infection, we believe that the mode of infection is a critical factor in disease development. IN or IO infections are performed using a nozzle that produces water mist and allows the virus containing fluid to enter the body cavity in an atomised form, which ensures effective and consistent infection.” Should be placed in line 346.

-        Line 349-351. Is this statement supported by your data “this indicates that the challenged virus replicates rapidly in vivo, not allowing for an immune response, which may not accurately replicate natural infection.” This is total speculation since the authors did not look at the immune response in any case and the natural infection develop immune response, most probably the cascade of cytokines (immune response) I what kills the animal.

-        Line 354-355. Again, the statement is totally speculative and it could be many other theories why the IN resulted in rapid death. The authors can discuss other possibilities. “However, in this study, IN administration resulted in rapid death, which indicates that an immune response did not occur.”

-        I would like to strongly recommend the authors to discuss the reproducibility of the challenge model IM alternative routes.

-        The authors should discuss why the different routes performed the same. It would have been expected differences when using up to 100 times higher doses of virus.

Reviewer 2 Report

I reviewed the manuscript entitled “Comparison of genotype II African swine fever virus strain SY18 challenge models for the evaluation of newly developed vaccine candidates”. In this study authors present a comprehensive pathogenesis analysis about the isolate SY18 of ASFV using diverse inoculation routes.

Overall, I think that the information generated from this study can be valuable to support the development of standard protocols intended to evaluate vaccine candidates of ASFV. However, I consider that the study has different issues that must be corrected before being considered for publication.

Based on the title of this study, in my opinion the main issue is the lack of animal experiments evaluating the performance of a vaccine candidate using the multiple inoculation routes proposed in this study. In this context, I suggest the authors to modify the title of their study for something more accurate like: “A comprehensive pathogenies study of the genotype II African swine fever virus strain SY18 using diverse inoculation routes”.  Based on the above, the information about the models of evaluation of newly developed vaccine candidates may be an aspect to be considered in the discussion section, and the study should be presented as a pathogenesis study.

Another issue in this manuscript is the introduction section. In my opinion, it seems to be out of context. In this sense, more information from previous studies regarding the pathogenesis of ASFV should be included in this section. I think the first part of the discussion may be used to improve the introduction section (lines 285-299).  This information will help to support the main goal of this study.

Also, I suggest avoiding personal communications. Information in this section and in the other sections of this manuscript should be supported with a reference. For example, line 67: “our findings and conversations with other researchers” should be substituted for published information.

Reviewer 3 Report

ASF is without any doubt the number on threat for the swine industry, negatively affecting global trading.

Investment in ASF research has been mostly neglected, contributing to explain the lack of efficient and safe treatments and vaccines for global use. This reality has recently changed and the efforts made are starting to render positive fruits.

On this regard, I consider that the work here presented (mostly reduced to tables 2-4 and figures 1-3) shows relevant data that deserves being published after carefully revision. 

Despite the data obtained deserved being shared with the scientific community, authors should concentrate their efforts to describe the different ASFV challenge models developed and the results obtained using NAÏVE PIGS, leaving for the discussion any claim about vaccination. In fact only one sentence in the whole article address this issue and is described as data not shown (Ll 296-300: During the development of our live vector vaccine, it was found that some recombinant virus candidate strains could not resist intramuscular challenge after immunisation, but could resist oral challenge (data not shown). Therefore, the development of alternative routes of infection is necessary to assess the immunoprotective effects of vaccines).

On this regard I strongly recommend starting by eliminating from the title any reference to vaccines. “ Comparison of genotype II African swine fever virus strain SY18 challenge models” might better reflect the reality than the original title. This is the best example I can find to illustrate that the entire manuscript should be revised and changed being more cautious and leaving for the discussion claims that are not backup with results. Time should confirm the suitability of these models to evaluate the efficacy of future vaccines.

Besides this, my other major concern is the poor methodological description of the challenge models. I consider ESSENTIAL to revise the corresponding section precisely describing the device use to nebulize the virus using the intranasal and the oral route, the volumes used, the immobilization method and any other detail ensuring its experimental reproduction by any other researcher.

Together with these crucial changes, I would appreciate providing responses to the following specific concerns:

1-Ll66  Based on our findings and conversations with other researchers, the dose of the challenge virus used for the gene-deleted ASF vaccine is usually 102.0 50% hemadsorbing doses (HAD50) to 104.0 HAD50, and this dose is too high for the efficacy evaluation of a subunit vaccine, which contains only some components of the ASFV. In addition, challenge methods significantly impact the induced immune response, and subunit vaccines usually induce an immune response that is resistant to oral rather than intramuscular challenges. In this study, using established

This paragraph from the introduction needs clarification or being moved to the discussion, since this does not sound as a scientific rational to justify the work performed. I insist that the data is worthy by itself without any reference to vaccination- A better understanding of the transmission of the virulent ASFV strain justifies the work here performed. The uses in the future to evaluate vaccines has to be demonstrated (in fact. I do not agree with the paragraph, as written)

2-I miss the use of no-vaccinated CONTROL PIGS (overall to compare pathological findings, clinical scoring etc…). This needs at least a justification

3- There are many tables and Figures with nom relevant or redundant and non-novel data. I would reduce them to tables 2-4 and figures 1-3.

4- CAREFUL!!!. Ll28-29 “Pigs that survived until day 28 were also euthanized

Is there any pig surviving until day 28? I could not see that reflected in the data. This is extremely important because as you also mentioned, many authors claim that with the exception of the IM challenge, no other ASFV challenge method is that reproducible. THIS IS IN FACT ONE OF THE STRONGEST POINTS OF THE PAPER; the mathematical reproducibility of the IN and oral challenges and as such will be judge in the future (it has to be reproducible by any researcher!!!)

5-Ll 134-137 “Oral and anal swabs were collected daily, at 10 am, from the day of inoculation with ASFV SY18. Concurrently, rectal temperatures were measured. To avoid interference factors, such as internal haemorrhage and pig stress, blood was not collected after inoculation.

I really miss the data of virus detection in blood (key parameter), so I recommend a better or more precise justification. Bleeding is invasive but immobilizing pigs with restrainers is (at least in my experience) the most stressful action for the pigs.

6- I recommend more carefully describing the data shown in the tables and the figures (not easy to follow in occasions… perhaps rephrasing in occasions)

7- Table 4 (change IM40 by I040) Mistake?

8-Figure 1. Careful!! the heads of each graph are wrong (IM200 in all???)

what it means the dotted line at the level of 70%-???

9-Revise the headings of the result section… Some poorly descriptive…

10- I would refer always to rectal and nasal swabs (the term “anal” sounds bad and referring to tonsil secretion might be a wrong statement, for example).Comparison between rectal and nasal swab provides surprising results that might contradict previous results, so they deserve a more careful description.

11- Better describe how much tissue is collected, processed… since the ASFV genome detection will depend on the amount of tissue used (in grams? 1ml of tissue means nothing)

Round 2

Reviewer 1 Report

The authors made an effort to correct the manuscript and it is greatly improved. I am overall happy with the corrections and the answers to my concerns.  I only have some extra comments and corrections for the authors to consider. 

-        Line 131. 1-2 mL should not be correct since all the above text is stating 2mL, please correct.

-       Line 262. I would suggest to the authors to modify the sentence “ASFV p72 genes were quantitatively…” and write the actual name of the gene B646L, p72 is the name of the protein.

-       Line 364. The amount of material in the rectal swab could be more.

-       Line 364. Minor differences do not affect our judgment of the infection characteristics of the virus.” I think what the authors are trying to say is that the differences were very small and not statistically significant.

-       Line 64. “An important reason for this result is that the current method of evaluating ASF vaccines, usually an intramuscular injection of highly virulent strains, is an unnatural model for vaccine evaluation.”. This is speculation and it’s not supported by any reference. We don’t know the reason and it’s very bold to assume the route of infection is the key to the subunit vaccine success.

-       Line 186. There is no group IO 500, I think the authors meant to write IO5000.

-       Line 191. There is no table 4.

-       Paragraph “In the IM200 group, the time between fever and death was 4, 3, and 5 days, whereas, in the IN and IO groups, the time from fever to euthanasia was 3–6 days. In addition, disease duration time(the time between the onset of symptoms and death or euthanasia ) generally increased in the IN and IO experimental groups (4.6 and 4.9 days, respectively) compared to that of the IM control group (4.0 days). However, the difference between the groups was not significant (Table 2, Duration).” The authors have to be consistent in how they show the results, if it’s the average of days or the range of days or mentioning every day. It is confusing for the reader. The authors should specify what the disease duration is, since fever is one symptom, what are the “other” symptoms to consider for this parameter. The whole paragraph needs review.

-       Paragraph “The overall survival time of the IO group was significantly longer than that of the IM group and IO group (about 18 days), among which the IO5000 subgroup was slightly shorter than the other three dose subgroups. The survival time of individuals in the lowest dose subgroup (IO40) was significantly different (> 6 days). (Table 2, Total days survival).” First, I think the authors meant to write IN where underlined. Second, is the significance of more than 6 days comparing low dose and high dose of the same group or comparing between groups. Either way, it needs correction.

-       Line 279. There is no table 5.

-       Line 340. The ASFV vaccine strain do not confer resistance to intramuscular challenge. The virus is not resistant or not resistant, the vaccine confers the resistance or the lack of the same.

- I would strongly recommend the authors to discuss or at least mention how it's possible not to detect virus in nasal swabs after intranasal injection of virus at time 0. 

Reviewer 2 Report

I like to thank the authors for their responses to my concerns. At this point, I don't have additional concerns about this study. 

Author Response

We are incredibly grateful to Reviewer #2 for the constructive comments and suggestions.

Reviewer 3 Report

I thank you the Authors for their careful revision

I consider that the manuscript is suitable for publication

Author Response

We are incredibly grateful to Reviewer #3 for the constructive comments and suggestions.